# Robot-Assisted Approach to Diabetes Care Consultations: Enhancing Patient Engagement and Identifying Therapeutic Issues

**DOI:** 10.3390/medicina61020352

**Published:** 2025-02-17

**Authors:** Yuya Asada, Tomomi Horiguchi, Kunimasa Yagi, Mako Komatsu, Ayaka Yamashita, Ren Ueta, Naoto Yamaaki, Mikifumi Shikida, Shuichi Nishio, Michiko Inagaki

**Affiliations:** 1Faculty of Health Sciences, Institute of Medical, Pharmaceutical and Health Sciences, Kanazawa University, Kanazawa 920-1192, Japan; y-asada@staff.kanazawa-u.ac.jp (Y.A.); horiguchi@mhs.mp.kanazawa-u.ac.jp (T.H.); ja9xbh@yahoo.co.jp (M.I.); 2Department of Internal Medicine, Kanazawa Medical University Hospital, Uchinada 920-0293, Japan; 3First Department of Internal Medicine, Toyama University Hospital, Toyama 930-0152, Japan; 4School of Informatics, Kochi University of Technology, Kochi 780-8515, Japan; komatsu.21151@gmail.com (M.K.); 260373c@ugs.kochi-tech.ac.jp (A.Y.); 285089e@gs.kochi-tech.ac.jp (R.U.); shikida.mikifumi@kochi-tech.ac.jp (M.S.); 5Isobe Clinic, Ichihara 290-0511, Japan; isobe_dm_clinic@yahoo.co.jp; 6Institute for Open and Transdisciplinary Research Initiatives, Osaka University, Osaka 565-0871, Japan; nishio@botransfer.org

**Keywords:** type 2 diabetes, communication robot, treatment behaviour, diabetes care consultation, response phrase

## Abstract

*Background and Objectives*: Diabetes is a rapidly increasing global health challenge compounded by a critical shortage of diabetes care and education specialists. Robot-assisted diabetes care offers a cost-effective and scalable alternative to traditional methods such as training and dispatching human experts. This pilot study aimed to evaluate the feasibility of using robots for diabetes care consultations by examining their ability to elicit meaningful patient feedback, identify therapeutic issues, and assess their potential as substitutes for human specialists. *Materials and Methods*: A robot-assisted consultation programme was developed by selecting an appropriate robot, designing the programme content, and tailoring back-channel communication elements. Experienced diabetes care nurses operated the robot during the consultations. Patient feedback was collected through a 17-item questionnaire using a five-point Likert scale (evaluating functionality, impressions, and effects). Additionally, a five-item questionnaire was used to assess whether the programme helped patients reflect on the key therapeutic domains of diabetes knowledge, diet, exercise, medications, and blood glucose control. *Results*: This study included 32 participants (22 males; mean age, 69.7 ± 12.6 years; mean HbA1c, 7.2 ± 1.0%). None of the participants reported any discomfort during the consultation. Sixteen of the seventeen feedback items scored above the median of 3, as did all five therapeutic reflection items. The interview content analysis revealed the programme’s ability to differentiate patients facing issues in treatment compliance from those effectively managing their condition. Robots can elicit valuable patient narratives like human specialists. *Conclusions*: The results of this pilot study support the feasibility of robot-assisted diabetes care to assist human experts. Future research should explore the programme’s application with healthcare professionals with limited experience in diabetes care, further demonstrating its scalability and utility in diverse healthcare settings.

## 1. Introduction

The prevalence of diabetes has been increasing rapidly worldwide [1], posing a significant challenge to healthcare systems. This escalating prevalence, combined with the global shortage of diabetes care and education specialists, highlights the urgent need for innovative solutions. Current approaches to addressing this shortage—such as training and dispatching human specialists—are resource-intensive in both time and cost. Therefore, alternative strategies that are cost-effective, scalable, and capable of rapid deployment are imperative.

Robot-assisted diabetes care is a promising approach in this context. Communication robots can be deployed swiftly and economically, potentially alleviating the burden on human specialists. Robots have demonstrated their utility in healthcare-related fields, including promoting physical activity in middle-aged and older adults [2], supporting mental health [3], improving cognitive function [4], rehabilitation for patients with hemiplegia [5], and mental support for patients with blood disorders [6]. Recent reports suggest that some individuals have more comfortable interactions with robots than with human healthcare professionals [7]. However, the applications of communication robots in diabetes care remain unexplored.

To address this gap, we developed a robot-assisted diabetes care consultation programme (hereafter referred to as “the programme”) to facilitate patient–professional interactions, identify therapeutic issues, and enhance patient outcomes. Previous studies have shown the potential of such programmes to improve glycaemic control before and after implementation [8]. Unlike conventional consultations, in which patients often experience hesitation or tension, robot-assisted consultations may provide a less intimate environment, thereby encouraging patients to share their concerns more openly.

Given this background, this study aimed to evaluate the feasibility of the programme by documenting its development process, assessing patient feedback, and determining whether healthcare professionals could effectively identify patients’ therapeutic issues. By exploring the role of robots in diabetes care, this study seeks to contribute to resolving the global shortage of diabetes care specialists and to improving patient outcomes.

## 2. Methods

### 2.1. Study Design and Ethical Issues

All procedures in this cross-sectional observational study followed the ethical standards of the responsible committee on human experimentation and the Helsinki Declaration of 1964, as well as its later amendments. The study protocol was approved by the Ethics Committee of Osaka University (IRB# R4-21). Written informed consent was obtained from all participants after they were informed that they could opt out at any time without penalty.

### 2.2. Study Participants

A diagnosis of diabetes was established based on the criteria outlined by the American Diabetes Association and the Japan Diabetes Society: HbA1c levels ≥ 6.5% (National Glycohaemoglobin Standardisation Programme), fasting blood glucose concentration ≥ 126 mg/dL (7.0 mmol/L), random blood glucose concentration ≥ 200 mg/dL, or current use of medications for diabetes [9,10]. The exclusion criteria were as follows: (i) type 1 diabetes, (ii) secondary diabetes, (iii) refractory malignant diseases, (iv) dependency on haemodialysis, (v) renal dysfunction with serum creatinine levels > 2.5 mg/dL, (vi) symptomatic coronary artery disease or percutaneous coronary intervention within the past year, (vii) severe hepatic dysfunction (Child–Pugh score ≥ 10), and (viii) patients who had changed their prescriptions within 2 months prior to the interview.

This study was conducted on patients with type 2 diabetes attending a private clinic in Japan. Consecutive cases were approached, and those who provided informed consent participated in this study. Given that this study targeted patients regularly attending a private clinic, it was anticipated that many would have relatively stable glycaemic control and fewer complications. To ensure the homogeneity of the study population, the exclusion criteria above were strictly applied.

Regarding the basic attributes of the subjects, information was collected on age, sex, HbA1c, presence or absence of injectable medication use, and presence or absence of complications.

### 2.3. Study Period

The study period was from November 2021 to October 2022.

### 2.4. Programme Creation

#### 2.4.1. Selection of the Robot

In this study, we used SHARP’s RoBoHoN-lite (SHARP Corporation, Osaka, Japan). We selected this robot for the following reasons: First, it has a rounded humanoid form, which gives it a charming appearance. Second, its size is convenient for portability because it can fit in one’s hand. Third, it is relatively inexpensive for robots, costing approximately JPY 100,000 (USD 154). Fourth, a previous study investigated the possibility of using this robot as a remote-control system [11].

#### 2.4.2. Programme Content

##### Creation of an Introduction

We created an introductory scenario to reduce the patient’s tension and clarify the patient’s behavioural role towards the robot.

##### Creation of Questions to Ascertain Treatment Behaviours

Nightingale [12] stated that humans should be viewed as an integrated entity in which the body, mind, and environment are interrelated. To ensure that these three aspects were depicted as a whole, multiple questions were constructed to capture the overall knowledge, acceptance, implementation level, and support system surrounding a specific health behaviour. Therefore, questions were created to assess whether patients with diabetes engage in the necessary behaviours for diabetes control, such as adjusting their environment, implementing desirable psychological and behavioural practices, and maintaining constructive relationships with their general practitioner (GP) and significant others. These questions were extracted based on guidelines for diabetes care [13]. Additionally, the content implemented by the researcher, a nurse with extensive experience in diabetes consultations, was also referenced. After creation, the questions were tested with three patients to determine whether they were understandable, confirm their relevance to the objectives, and check the appropriateness of the order of the questions, leading to revisions of the scenario.

Treatment Behaviours (Diet, Exercise) (Table 1)

**Table 1 medicina-61-00352-t001:** RoBoHoN interview items and assessment content.

Interview Items		Assessment Content
Treatment Behaviours	Diet	Whether there are any awkward situations with family regarding meals
		Satisfaction with meals
		Relationship between diet and blood glucose levels
		Self-evaluation of how well things are going
		Whether family members give compliments
		Whether you eat anything other than meals
	Exercise	Whether you share details about your exercise, such as the type and duration, with your doctor
		Selecting and performing exercise based on your physical condition
		Exercise on rainy days
		Whether you have any questions about exercise
		Whether you did any exercise in your youth
Treatment Knowledge	Foot care	Whether you are familiar with the term “foot care”
		Knowledge of foot care methods, such as nail trimming
	Sick day management	Whether you are familiar with the term “sick day management”
		Whether you have ever sought a medical consultation on days other than your regular check-up days
	Complications	Whether you want detailed information about complications and how to prevent them
		The level of ability to explain the complications you know
Diabetes Acceptance	Aetiology	Perception of the cause of diabetes
	Whether there is any disadvantage or burden in daily life	Whether the thought of having diabetes itself feels like a burden
		Changes in lifestyle following the diagnosis
		Whether you feel you have lost out because of your diabetes
	Expectations towards the people around you	Whether you have expectations towards healthcare professionals
		Whether you have expectations towards family
		Whether you have expectations towards RoBoHoN

The content related to diet was composed of actual dietary intake, its relationship with blood glucose levels, snacking habits, evaluations, satisfaction with the current diet, and the relationship with family and support staff regarding meals.

The content related to exercise was structured to include the type and duration of exercise being performed, the implementation status according to the weather and physical condition, questions regarding exercise, and the status of consultations with GP.

b.Treatment Knowledge (Table 1)

This section comprised content related to foot care, sick day management, and complications.

c.Diabetes Acceptance (Table 1)

This section was structured to include perceptions of the causes of diabetes onset, disadvantages and burdens of living with diabetes, and expectations regarding the surrounding environment (healthcare professionals and family members).

#### 2.4.3. Creation of Response Phrases Emitted by the Robot During Interviews

Response phrases were created to deepen the interviews and ease speech. The creation of these phrases was based on counselling theories [14], interpersonal communication theories [15], and learning theories [16]. To determine the appropriateness of the content, trials were conducted with three patients, and the operator observed and confirmed whether the response phrases interrupted the patient’s speech, whether the content developed to a certain depth, and whether the content caused unpleasant reactions. Based on these findings, the response phrases were finalised. The system was designed to allow the robot to select and vocalise responses that the operator judged to be appropriate according to the patient’s responses to questions about treatment behaviour and the content of the patient’s speech.

#### 2.4.4. Interview Procedure

An interview was conducted with one subject and a robot operator. The robot was operated by nurses experienced in diabetes consultations. The robot operator sat slightly apart from the subject in the same room and operated the robot. The operator did not explicitly inform the subject that the robot was being operated by a person in the same room during the interviews. The interview was recorded using a camera and microphone integrated into the robot, enabling a review of the interview content.

### 2.5. Evaluation Method of the Programme

#### 2.5.1. Participant Evaluations

After robot-mediated interviews, a questionnaire was conducted to evaluate the robot. The evaluation included three aspects of the robot itself: “Functionality of the RoBoHoN”, “Impression of the RoBoHoN”, and “Effects of the RoBoHoN”. Additionally, participants were asked about the programme content: “Did the programme help you reflect on your diabetes?”. The responses were rated on a five-point Likert scale, from “1. Strongly disagree” to “5. Strongly agree”. We checked the internal consistency of these items and obtained a Cronbach’s alpha coefficient of 0.87. Since a Cronbach’s alpha coefficient of ≥0.7 is considered acceptable, it was determined that the items in this study were valid.

#### 2.5.2. Criteria for Assessing the Patient’s Issues in Diabetes Care

Through conversations with the robot, the patients described three aspects of issues in diabetes care: (1) treatment behaviours (e.g., diet and exercise), (2) treatment knowledge, and (3) diabetes acceptance. Criteria were established to assess the patients’ therapeutic issues based on these three aspects. After the interviews, assessments were conducted by two nurses with experience in diabetes consultation who reviewed the interviews.

##### Treatment Behaviours (e.g., Diet, Exercise) (Table 1)

The criteria for assessing diet and exercise were established from three perspectives: knowledge, behaviour, and support from others. The scores for this section range from 3 to 9 points.

  (i)Knowledge: 3 levels (1 point: No knowledge, 2 points: Aware of precautions, 3 points: Have basic knowledge) (ii)Behaviour: 3 levels (1 point: Not concerned, 2 points: Only pay attention, 3 points: Practising basic principles)(iii)Support from others: 3 levels (1 point: No concept of cooperation from others, 2 points: Feel that others cooperate but cannot talk about specifics, 3 points: Can discuss specifics regarding cooperation)

##### Treatment Knowledge (Table 1)

These criteria were based on the presence or absence of treatment knowledge, with three assessment levels: 1 point, never heard of it; 2 points, heard of it but unable to explain; and 3 points, knowledgeable and able to explain. The scores for this section ranged from 3 to 9 points.

##### Diabetes Acceptance (Table 1)

The criteria for diabetes acceptance were as follows: scores for this section ranged from 3 to 6 points.

  (i)Cause of onset: 2 levels (1 point: Do not know, 2 points: Have an idea). (ii)Burden or disadvantage in daily life: 2 levels (1 point: None, 2 points: Present).(iii)Expectations from others: 2 levels (1 point: None, 2 points: Present).

### 2.6. Statistical Analysis

The sample size was determined using Lehr’s formula. Assuming an expected difference in questionnaire score of 1 and a standard deviation of 0.7, the effect size was estimated to be 1.4. Consequently, the required sample size was calculated as 32 (16 × 1.4^2^), with a target power of 80% and a significance level of 0.05.

Continuous variables are expressed as mean ± standard deviation (SD) and median values, while categorical variables are reported as counts and percentages. For questionnaire responses, 95% confidence intervals (CIs) were calculated. The Kolmogorov–Smirnov test was used to analyse the cumulative relative frequency distributions.

Statistical analyses were conducted using Microsoft Excel (Microsoft Corporation, Redmond, WA, USA), R version 4.3.0 (R Foundation for Statistical Computing, Vienna, Austria), GUI version 1.79, RStudio version 2024.09.1+394 (Boston, MA, USA) on an Apple Macintosh computer (Apple, Cupertino, CA, USA), and SPSS Statistics version 27 (IBM, Chicago, IL, USA).

## 3. Results

### 3.1. Baseline Characteristics of the Participants (Table 2)

In total, interviews were conducted with 32 participants (22 males [68.8%]; mean age, 69.7 ± 12.6 years). All participants were asked a series of questions and responses were recorded for each question. The average HbA1c level was 7.2 ± 1.0%, and a limited number of participants were treated with insulin injections (28.1%) and had diabetic complications (15.6%).

**Table 2 medicina-61-00352-t002:** Baseline characteristics of the participants. (n = 32).

Variables	Mean ± SD
Male gender, (%)	22 (68.8)
Age, years old	69.7 ± 12.6
HbA1c, %	7.2 ± 1.0
Treated with insulin injections, (%)	9 (28.1)
With diabetic complications, (%)	5 (15.6)

### 3.2. Created Programme

#### 3.2.1. Introduction

In the introduction, the expected role behaviour was: “I will ask you questions about your diet, exercise, and treatment, so please answer them”.

#### 3.2.2. Questions Regarding Treatment Behaviours (Table 3)

Regarding treatment behaviours, the questions about diet included the following: “Are you satisfied with your current diet? Can you tell me if you’re satisfied?” and “ Do you ever think that your diet is a cause of your fluctuating blood glucose levels?”. For family relationships related to meals, the question was: “Do you ever have awkward moments with your family because of meals? For example, are you often told you eat too much?”.

Regarding exercise, the questions included the following: “Do you choose and perform exercises based on your condition each day? What do you do when it rains? How do you exercise on rainy days?”. The question about exercise-related concerns was “If you have any questions about exercise, please feel free to ask me”.

Regarding treatment knowledge, the questions about foot care included the following: “Are you familiar with foot care?” and “If you have any trouble with foot care, such as cutting your nails, please tell me”. Regarding sick day management, the questions were: “Do you know what a ’sick day management’ is?” and “If you have ever had the experience of wanting to see a doctor outside of your routine appointments, please share it with me”.

Regarding complications, the questions included the following: “Would you like to know more details about the types of complications and their prevention?” and “Can you tell me about any complications you’re aware of?”. In addition, questions regarding oral medication and insulin therapy were added to the questionnaire.

Regarding diabetes acceptance, the questions were “Could you let me know if you feel burdened just because you have diabetes?” and “Please tell me what you expect from your healthcare professional or family when you go through treatment for diabetes”.

All questions and scenarios are presented in Table 3. No participants exhibited unpleasant reactions throughout the programme, and all were able to respond to the questions.

**Table 3 medicina-61-00352-t003:** Questions regarding treatment behaviours.

**Introduction**	I will ask you questions about your diet, exercise, and treatment, so please answer them.
	Please rest assured that your responses will not be shared with anyone without your permission.
	I will come up with useful advice regarding subsequent treatment based on what you tell me.
	I will discuss matters with your GP as necessary.
	Thank you in advance for your time and cooperation.
**Treatment** **Behaviours** **(Diet and Exercise)**	I will ask you about your daily life, so please share your thoughts.
	First, I will ask about your diet.
	Do you ever have awkward moments with your family because of meals? For example, are you often told you eat too much?
	Are you satisfied with your current diet? Can you tell me if you’re satisfied?
	Do you ever think that your diet is a cause of your fluctuating blood glucose levels?
	Please tell me whether your family praises you for your efforts in following your dietary regimen.
	Do you think your current diet is working well?
	Think back to yesterday’s meals and tell me what you ate.
	Is that generally similar to a normal meal for you?
	Please let me know if you eat anything in addition to your meals.
	That concludes the questions about your diet.
	Next, I will ask about exercise.
	Could you let me know if you communicate the types and duration of exercise you do to your primary physician.
	Do you choose and perform exercises based on your condition each day?
	What do you do when it rains? How do you exercise on rainy days?
	If you have any questions about exercise, please feel free to ask me.
	If there were any sports you played when you were younger, please share that with me.
	That concludes the questions about exercise.
	This wraps up the questions about your lifestyle.
**Treatment Knowledge**	From now on, I will ask questions about your treatment.
	Are you familiar with foot care?
	If you have any trouble with foot care, such as cutting your nails, please tell me.
	Do you know what a ’sick day management’ is?
	If you have ever had the experience of wanting to see a doctor outside of your routine appointments, please share it with me.
	Would you like to know more details about the types of complications and their prevention?
	Can you tell me about any complications you’re aware of?
	Do you feel that your insulin injections are going well?
	If you have any questions about insulin injections, please let me know.
	If you have any questions about the medication you are currently taking, please let me know.
**Diabetes** **Acceptance**	Could you tell me what you think is the cause of your diabetes?
	We are almost done with the questions. Please do your best to answer.
	Could you let me know if you feel burdened just because you have diabetes.
	If you feel that your lifestyle has changed since being diagnosed with diabetes, please share what has changed.
	If you have ever felt that having diabetes has caused you to lose out on something, please let me know.
	The following questions are about your expectations for healthcare professionals and your family.
	Please tell me what you expect from your healthcare professional or family when you go through treatment for diabetes.
	If you have any requests for me, the robot asking these questions, please let me know.
**Conclusion**	This is the last question.
	If there is anything else you would like to ask me, about anything other than the questions I asked today, feel free to ask.
	That concludes the questions. Thank you very much.

**GP, general practitioner.**

#### 3.2.3. Response Phrases Issued by the Robot During the Interview (Table 4)

The response phrases were designed as shown in Table 4. These included expressions of empathy such as “I see”, “Right”, “That’s good”, and “Yes, indeed”. To encourage the participants to elaborate further or reflect on their thoughts, phrases such as “Really?’, “Please tell me about your current situation in a little more detail”, and “It sounds as though you don’t see that as a good thing…” were used. Encouraging phrases included “Thank you for sharing this information with me” and phrases to motivate treatment behaviours, such as “That’s wonderful!” and “You’ve certainly been doing your best”. To encourage awareness of the healthcare team, phrases such as “I will discuss this with your GP” and “I will ask your GP in an indirect way” were included.

In total, 22 response phrases were developed. These phrases facilitated the flow of conversation, encouraged participants to provide more detailed responses, and helped maintain their engagement throughout the interviews. As the conversations were originally conducted in Japanese, the original Japanese phrases are presented with their English equivalents in Appendix A for reference.

**Table 4 medicina-61-00352-t004:** Response phrases by the RoBoHoN during Interviews.

I see.	Right.
That’s good.	Yes, indeed.
Is that so.	It really is.
I see it’s been on your mind as well.	Was this question difficult to answer?
Really?	It sounds as though you don’t see that as a good thing…
Please tell me about your current situation in a little more detail.	Oh wow, I never would have thought.
Did you ever wish you had more support?	Could you tell me why?
Indeed, it’s not just about you.	How do you cope in those situations?
Please tell me the specific amount, and other details.	Thank you for sharing this information with me.
That is wonderful!	You’ve certainly been doing your best.
I will discuss this with your GP.	I will ask your GP in an indirect way.

GP, general practitioner.

### 3.3. Evaluation by the Participants (Table 5)

The average score ranges for the evaluation of the communication robot are as follows: For “Functionality of RoBoHoN” (seven items), the average scores ranged from 3.45 to 4.61 points. For “Impression of RoBoHoN” (five items), the average scores ranged from 3.16 to 4.42 points. For “Effects of RoBoHoN” (five items), the average scores ranged from 2.81 to 3.90 points, with the item ’I thought it would make attending the clinic more enjoyable’ being the only one scoring below 3 points. Regarding the average scores for the evaluation of the programme content, the average scores for the five items of “Whether the programme led to a reflection on diabetes” ranged from 3.52 to 3.90.

**Table 5 medicina-61-00352-t005:** Evaluation of the programme by the participants. (n = 31 *).

Questionnaire Items		Strongly Disagree	Disagree	Neither Agree Nor Disagree	Somewhat Agree	Strongly Agree	Mean ± S.D.	95% CI
Functionality of RoBoHoN	RoBoHoN’s voice was easy to understand.	0 (0.0)	1 (3.2)	2 (6.5)	10 (32.3)	18 (58.1)	4.45 ± 0.77	4.17–4.73
	The speaking pace was just right.	0 (0.0)	0 (0.0)	3 (9.7)	6 (19.4)	22 (71.0)	4.61 ± 0.67	4.37–4.86
	I felt that I was listened to.	0 (0.0)	1 (3.2)	4 (12.9)	11 (35.5)	15 (48.4)	4.29 ± 0.82	3.99–4.59
	The timing of the questions felt natural.	0 (0.0)	0 (0.0)	8 (25.8)	12 (38.7)	11 (35.5)	4.10 ± 0.79	3.81–4.38
	The content of the questions was difficult. ^†^	0 (0.0)	5 (16.1)	12 (38.7)	9 (29.0)	5 (16.1)	3.45 ± 0.96	3.10–3.80
	The timing of the responses was natural.	0 (0.0)	0 (0.0)	11 (35.5)	11 (35.5)	9 (29.0)	3.94 ± 0.81	3.64–4.23
	The content of the responses was natural.	0 (0.0)	1 (3.2)	9 (29.0)	12 (38.7)	9 (29.0)	3.94± 0.85	3.62–4.25
Impression of RoBoHoN	It was cute.	0 (0.0)	1 (3.2)	6 (19.4)	9 (29.0)	15 (48.4)	4.23 ± 0.88	3.90–4.55
	I had a negative impression (such as fear or coldness). ^†^	0 (0.0)	1 (3.2)	3 (9.7)	9 (29.0)	18(58.1)	4.42 ± 0.81	4.12–4.72
	I developed a sense of attachment while talking.	4 (12.9)	3 (9.7)	13 (41.9)	5 (16.1)	6 (19.4)	3.19 ± 1.25	2.74–3.65
	I felt a sense of familiarity in the way the robot spoke.	2 (6.5)	3 (9.7)	5 (16.1)	11 (35.5)	10 (32.3)	3.77 ± 1.20	3.33–4.22
	I wanted to talk more.	2 (6.5)	3 (9.7)	16 (51.6)	8 (25.8)	2 (6.5)	3.16 ± 0.93	2.82–3.50
Effects of RoBoHoN	I felt that it was easier to talk than when speaking with a healthcare professional.	2 (6.5)	3 (9.7)	16 (51.6)	6 (19.4)	4 (12.9)	3.23 ± 1.02	2.85–3.60
	It was easier to talk about things that are difficult to say compared with speaking with a healthcare professional.	0 (0.0)	6 (19.4)	13 (41.9)	7 (22.6)	5 (16.1)	3.35 ± 0.99	2.99–3.72
	I felt more pressure of being “checked” compared with talking with a healthcare professional. ^†^	0 (0.0)	2 (6.5)	8 (25.8)	12 (38.7)	9 (29.0)	3.90 ± 0.91	3.57–4.24
	The childish way of speaking made me feel reluctant to talk about private matters. ^†^	1 (3.2)	2 (6.5)	8 (25.8)	8 (25.8)	12 (38.7)	3.90 ± 1.11	3.50–4.31
	I thought it would make attending the clinic more enjoyable.	3 (9.7)	5 (16.1)	20 (64.5)	1 (3.2)	2 (6.5)	2.81 ± 0.91	2.47–3.14
Reflection on diabetes	By having this conversation, I was able to reflect on my knowledge of diabetes.	0 (0.0)	1 (3.2)	8 (25.8)	18 (58.1)	4 (12.9)	3.81 ± 0.70	3.55–4.06
	By having this conversation, I was able to reflect on my diet.	0 (0.0)	2 (6.5)	10 (32.3)	10 (32.3)	9 (29.0)	3.84 ± 0.93	3.50–4.18
	By having this conversation, I was able to reflect on my exercise routine.	0 (0.0)	1 (3.2)	10 (32.3)	11 (35.5)	9 (29.0)	3.90 ± 0.87	3.58–4.22
	By having this conversation, I was able to reflect on my medication (oral medication or insulin).	1 (3.2)	3 (9.7)	12 (38.7)	9 (29.0)	6 (19.4)	3.52 ± 1.03	3.14–3.89
	By having this conversation, I was able to reflect on my blood glucose control.	1 (3.2)	1 (3.2)	7 (22.6)	13 (41.9)	9 (29.0)	3.90 ± 0.98	3.54–4.26

* One participant was unable to complete the questionnaire for the robot evaluation; ^†^ Calculated as a reverse-scored item; CI, confidential interval; S.D., standard deviation.

### 3.4. Distribution of Scores for Treatment Behaviours (Figure 1a,b)

The total score for diet showed that many participants scored 3 (n = 8) or 5 points (n = 9). A score of 3 points indicated a significant concern with diet; about 25% of the participants fell into this category. Furthermore, because the score range for diet was distributed from 3 to 8 points, this suggests that the assessment was tailored to each participant’s individual situation.

For exercise, the most common total score was 3 points (n = 11). A score of 3 points indicated a concern with exercise; approximately 35% of the participants fell into this category. The score range for exercise was from 3 to 9 points, which also enabled an assessment tailored to the individual circumstances of each participant.

### 3.5. Distribution of Scores for Treatment Knowledge (Figure 1c)

The total score for treatment knowledge was as follows: the most common score was 4 points (n = 15), followed by 3 points (n = 9). The minimum score was 3 points, indicating a lack of knowledge, whereas a score of 4 points suggested that the participant had heard of at least one of the three topics. Therefore, approximately 70% of the patients lacked sufficient knowledge. With scores ranging from 3 to 9 points, this suggests that the assessment of treatment knowledge was also tailored to each participant’s individual situation.

### 3.6. Distribution of Scores for Diabetes Acceptance (Figure 1d)

The most common total score for acceptance of diabetes was 4 points (n = 15). The score for diabetes acceptance ranged from 3 to 6 points, indicating that the assessment of diabetes acceptance was adapted to the individual circumstances of each participant.

**Figure 1 medicina-61-00352-f001:**
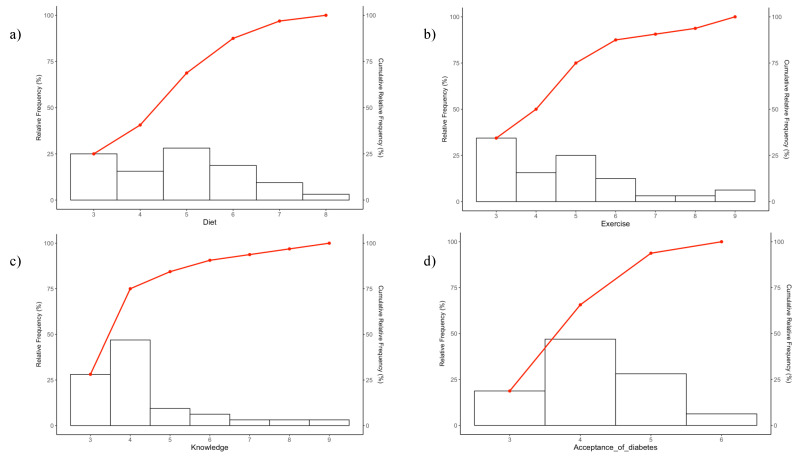
Histograms and cumulative relative frequency graphs depicting the distribution of scores related to treatment behaviours: (**a**) diet, (**b**) exercise, (**c**) treatment knowledge, and (**d**) diabetes acceptance.

### 3.7. Differences in the Score Distribution of Treatment Behaviours

The Kolmogorov–Smirnov test was used to examine differences in score distributions across treatment behaviours, treatment knowledge, and diabetes acceptance. The analysis revealed a statistically significant difference between diet and treatment knowledge (*p* = 0.016) (Table 6).

## 4. Discussion

### 4.1. Evaluating the Feasibility of Robot-Assisted Therapeutic Consultations for Diabetes Care

Effective therapeutic consultations for diabetes management require patients to communicate their experiences openly with healthcare professionals without causing discomfort or hesitation. This study evaluated the feasibility of a robot-assisted consultation programme by examining three key aspects: the suitability of the selected robot model, the design and content of the questions and responses, and the programme’s ability to assess treatment behaviours.

### 4.2. The Selected Robot

This study highlights the significance of human–robot interaction in healthcare, where both entities can complement each other’s strengths while enhancing overall engagement. Patient feedback indicated that they found it easier to discuss sensitive topics with the robot than with healthcare professionals. Previous research has explored the potential of tablet-assisted medical guidance [17]. However, given Japan’s rapidly ageing population and evolving family dynamics, many individuals now have fewer opportunities for human support and seek alternative means of interaction.

Orem’s self-care theory emphasises that “caring” is a fundamental human need [18]. The fact that the patients described the robot as “cute” and developed a sense of attachment suggests that interacting with a robot perceived as non-threatening may help reduce patient anxiety. However, only a small proportion of the participants reported that the robot enhanced their motivation to visit the clinic, indicating that it was not a primary factor in encouraging clinic attendance.

Rather than allowing the robot to operate independently, healthcare professionals actively selected and administered questions and responses. This approach was designed to foster a collaborative dynamic among patients, healthcare professionals, and robots, engaging all parties in meaningful therapeutic discussions. While the extent to which a true sense of camaraderie was achieved remains unclear, this study successfully demonstrated the feasibility of integrating robots into diabetes care consultations as interactive facilitators rather than passive tools.

The financial burden associated with robotic implementation is a recognised concern [19,20]. However, the RoBoHoN used in this study is relatively affordable compared to other humanoid robots. Moreover, its potential for long-term use may help offset initial costs, ultimately leading to cost savings when compared to labour expenses. Additionally, the installed programme allows for intuitive operation, requiring only word selection and a button press. Given these advantages, RoBoHoN demonstrates the potential for widespread adoption and sustainable integration into clinical practice.

### 4.3. Contents of the Questions and Responses

This programme was designed to facilitate patient-led discussions regarding knowledge, skills, and support systems related to compliance with treatment behaviours. The robot prompted open-ended discussions, asking, “Do you think your current diet is working well?” Multiple questions were developed to assess various dimensions of diabetes care, including knowledge, acceptance, implementation, and support structure. Most participants did not find the questions difficult to answer, and many reported that they could reflect on their understanding of diabetes management. These findings suggest that robot-facilitated dialogue successfully elicited patient narratives, demonstrating its potential as an interactive tool for diabetes care consultation.

An essential feature of this programme is the incorporation of backchannelling, a communication technique that encourages active conversation engagement. The 22 backchannelling phrases used in this study were carefully selected from among those commonly employed by experienced healthcare professionals during consultations. Patient evaluations indicated that very few participants found the timing or content of the backchannelling responses unnatural. In a previous study, a robot was used to confirm knowledge about diabetes, and a questionnaire was used to evaluate the findings [21]. Similarly, our findings highlight the positive reception of robotic interactions, particularly in terms of conversational fluency and engagement. Notably, robot-assisted backchannelling proved effective in sustaining patient narratives without disruption, even within the more complex framework of this study, in which patients were asked to discuss their medical treatment history.

These results indicate that the questions and backchannelling strategies developed in this programme can facilitate meaningful patient discussions from multiple perspectives. These elements are regarded as useful for identifying issues in patient management and for developing solutions.

### 4.4. Assessment of Issues in Treatment Behaviours

In this study, 30–40% of participants exhibited difficulties complying with dietary and exercise regimens, consistent with previous findings that reported 20–30% non-compliance rates in these behaviours [22].

Histogram analysis indicated that participants scored lower on treatment knowledge than on dietary behaviour. Regarding treatment knowledge, most participants demonstrated limited understanding. This finding is consistent with previous studies, which reported that nearly half of patients did not routinely inspect their feet [23] and that many had little or no awareness of the concept of “sick day management” [24]. The observed lack of recall in our study may be attributed to the participants’ relatively stable blood glucose control, absence of foot complications, and limited personal experience with acute illness episodes.

In assessing diabetes acceptance, factors such as understanding the cause of disease onset, perceived burden on daily life, and expectations from others were identified as key elements that could be explored through therapeutic consultations. The distribution of scores suggests that, even with RoBoHoN-assisted consultations, it was possible to facilitate meaningful discussions on these critical aspects of treatment behaviours. Furthermore, the patient narratives elicited through robotic interviews aligned with trends observed in previous studies, indicating that robot-assisted consultations can yield insights comparable to those obtained through clinician-led interactions.

Future research should focus on evaluating long-term changes in patient compliance and behaviour following robot-assisted consultations, as well as assessing the effectiveness of this approach in enhancing diabetes management outcomes.

### 4.5. Limitations of This Study

This study has several limitations due to its single-centre design and private practice setting. The study population comprised patients with relatively stable blood glucose levels and no severe complications. The findings were based on a single-point interview conducted with a robot. These factors inherently limit the generalisability of the results. Additionally, as the robot in this study was operated by an experienced diabetes care specialist, future research should assess whether similar outcomes can be achieved when operated by healthcare professionals with less experience in diabetes care consultation.

However, these limitations also reflect the reality of diabetes management, where interventions often target asymptomatic patients to promote behavioural changes. This study provides valuable insights into a forward-looking approach to diabetes care in which specialists leverage robotic assistance to enhance therapeutic interventions in real-life settings. Nevertheless, the potential role of robots in the management of patients with established complications remains an important issue and warrants further investigation.

Although the sample size of this study was limited, it was statistically sufficient to draw meaningful conclusions. Unlike large-scale studies where clinically insignificant but statistically significant differences may emerge, this study evaluated the practical benefits and applicability of robot-assisted consultations in diabetes care. While acknowledging its limitations, the findings offer important insights into the feasibility and effectiveness of this approach in a specific healthcare environment.

## 5. Conclusions

In this study, we developed a consultation programme for diabetes care using a communication robot (RoBoHoN) and evaluated its applicability for patients. The results indicate the feasibility of robot-assisted therapeutic consultations for diabetes care, potentially progressing to assist human experts.

## Figures and Tables

**Table 6 medicina-61-00352-t006:** Kolmogorov–Smirnov tests for the differences in the score distribution of treatment behaviours.

Comparison	KS Statistic	*p*-Value
Diet behaviour vs. Exercise behaviour	0.093	0.887
Diet behaviour vs. Treatment knowledge	0.34	0.016
Diet behaviour vs. Diabetes acceptance	0.25	0.116
Exercise behaviour vs. Treatment knowledge	0.25	0.118
Exercise behaviour vs. Diabetes acceptance	0.19	0.240
Treatment knowledge vs. Diabetes acceptance	0.094	0.891

KS Statistic, The Kolmogorov–Smirnov statistic (maximum difference between empirical cumulative distribution functions).

## Data Availability

The datasets used and analysed in this study are available from the corresponding author upon reasonable request.

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
