# Peer review of "Robot-Assisted Approach to Diabetes Care Consultations: Enhancing Patient Engagement and Identifying Therapeutic Issues"

_medicina, 2025, doi:10.3390/medicina61020352_

Round 1
Reviewer 1 Report
Comments and Suggestions for Authors
Review report: Robot-Assisted Approach to Diabetes Care Consultations: Enhancing Patient Engagement and Identifying Therapeutic Issues
· A brief summary: The authors have done a great job on this project. The study evaluated the feasibility of using robots in diabetes care consultations by examining their ability to elicit meaningful patient feedback, identify therapeutic challenges, and assess their potential as substitutes for human specialists. The paper contributes to the field by sharing knowledge on developing a robot-assisted consultation program, demonstrating its effectiveness in engaging patients and assessing their therapeutic behaviors, and exploring its potential to address the global shortage of diabetes care professionals. The study's strength lies in its comprehensive approach to program development, including careful robot selection, design of program content, and tailoring of communication elements. Furthermore, the study uses multiple evaluation methods, including patient feedback and assessment of therapeutic behaviors, providing a well-rounded view of the program's effectiveness.
· General concept comments:
o The manuscript is clear and relevant to the field of diabetes care and healthcare technology, and it is presented in a well-structured manner. However, I believe the part concerning statistical analysis in methods could be described in more detail.
o The cited references are appropriate/relevant, and most are within the last 5 years.
o The manuscript is scientifically sound, and the experimental design is appropriate for a pilot study. However, in my view, there are some limitations and areas where the scientific rigor could be improved. The methods section should be beefed up mainly with statistical processing.
§ For example, more details on how the sample size was estimated are needed. How was randomization handled? How were biases in sample selection handled?
§ The sample size of 32 participants is too small to generalize the results.
o Although the manuscript would provide a good foundation for reproducibility, especially after addressing the issues raised above, there are some areas where additional details could enhance the ability to replicate the study.
o The manuscript includes tables and figures that are appropriate and helpful in presenting the data. The statistical analysis is minimal, primarily consisting of descriptive statistics (mean values and standard deviations). No significance was tested. More advanced statistical analyses could have been employed to strengthen the findings and provide more insights into the data. In short, more detailed statistical analysis, including tests for significance or correlations between different variables, could enhance the robustness of the findings.
o The conclusions presented in the manuscript are broadly consistent with the evidence and arguments presented. However, the connection could be strengthened in some places, or additional caution in interpretation might be warranted.
§ For example, while the study demonstrates the feasibility of the robot-assisted program, the conclusion that it can serve as a 'viable substitute for human experts' may be somewhat overstated, given the study's limitations. The lack of a direct comparison with human-led consultations makes this claim difficult to substantiate fully.
§ The authors could more explicitly acknowledge the study's limitations, such as the small sample size and the relatively stable condition of the participants.
o The manuscript includes ethics and data availability statements are adequate.
o Finally, I believe this is a great research and should be accepted for publication after minor revisions.

Author Response
To Reviewer #1:
Review report: Robot-Assisted Approach to Diabetes Care Consultations: Enhancing Patient
Engagement and Identifying Therapeutic Issues
• A brief summary: The authors have done a great job on this project. The study evaluated the feasibility of using robots in diabetes care consultations by examining their ability to elicit meaningful patient feedback, identify therapeutic challenges, and assess their potential as substitutes for human specialists. The paper contributes to the field by sharing knowledge on developing a robot-assisted consultation program, demonstrating its effectiveness in engaging patients and assessing their therapeutic behaviors, and exploring its potential to address the global shortage of diabetes care professionals. The study's strength lies in its comprehensive approach to program development, including careful robot selection, design of program content, and tailoring of communication elements. Furthermore, the study uses multiple evaluation methods, including patient feedback and assessment of therapeutic behaviors, providing a well-rounded view of the program's effectiveness.
Thank you for your sincere comments. We appreciate them.
• General concept comments:
o The manuscript is clear and relevant to the field of diabetes care and healthcare technology, and it is presented in a well-structured manner. However, I believe the part concerning statistical analysis in methods could be described in more detail.
Thank you for your valuable feedback. As you pointed out, the statistical analysis section lacked sufficient detail. We have now provided additional information to clarify our methodology. Please let us know if further clarification is needed.
o The cited references are appropriate/relevant, and most are within the last 5 years.
Thank you for your comments. We appreciate them.
o The manuscript is scientifically sound, and the experimental design is appropriate for a pilot study. However, in my view, there are some limitations and areas where the scientific rigor could be improved. The methods section should be beefed up mainly with statistical processing.
Thank you for your comments. We acknowledge your point regarding the need for additional statistical details.
â–ª For example, more details on how the sample size was estimated are needed. How was randomization handled? How were biases in sample selection handled?
Thank you for your thoughtful comments. In response to your suggestion, we have added details on how the sample size was determined. Please let us know if any further clarifications are needed.
â–ª The sample size of 32 participants is too small to generalize the results.
Thank you for your thoughtful comments. We acknowledge that a sample size of 32 participants is relatively small. However, as we have now clarified in the manuscript, the sample size was determined to ensure statistical validity while maintaining practical feasibility. Additionally, as noted in the Limitations section of the Discussion, the moderate sample size helped avoid the risk of detecting statistically significant but clinically insignificant results. We appreciate your review and hope our revisions sufficiently address your concerns.
o Although the manuscript would provide a good foundation for reproducibility, especially after addressing the issues raised above, there are some areas where additional details could enhance the ability to replicate the study.
Thank you for your thoughtful comments. We appreciate your valuable feedback and acknowledge the importance of providing sufficient details to enhance the reproducibility of our study.
o The manuscript includes tables and figures that are appropriate and helpful in presenting the data. The statistical analysis is minimal, primarily consisting of descriptive statistics (mean values and standard deviations). No significance was tested. More advanced statistical analyses could have been employed to strengthen the findings and provide more insights into the data. In short, more detailed statistical analysis, including tests for significance or correlations between different variables, could enhance the robustness of the findings.
Thank you for your thoughtful comments. As you noted, our initial statistical analysis was primarily descriptive, reflecting the inherent limitations of a single-arm study. To enhance the robustness of our findings, we have incorporated inferential statistical methods where applicable. Specifically, we evaluated the 95% confidence intervals for the interview results, added cumulative relative frequency graphs to the histograms, and conducted assessments using the Kolmogorov-Smirnov test. We appreciate your careful review and hope these revisions address your concerns.
o The conclusions presented in the manuscript are broadly consistent with the evidence and arguments presented. However, the connection could be strengthened in some places, or additional caution in interpretation might be warranted.
Thank you for your thoughtful comments. We sincerely appreciate your insights.
â–ª For example, while the study demonstrates the feasibility of the robot assisted program, the conclusion that it can serve as a 'viable substitute for human experts' may be somewhat overstated, given the study's limitations. The lack of a direct comparison with human-led consultations makes this claim difficult to substantiate fully.
Thank you for your insightful comments. As you rightly pointed out, our initial wording may have overstated the conclusion in emphasizing the role of the robot-assisted program. We have carefully revised the conclusion and the corresponding abstract to reflect that the program can assist human experts rather than serve as a direct substitute. We appreciate your guidance and hope these revisions improve our findings' clarity and accuracy.
â–ª The authors could more explicitly acknowledge the study's limitations, such as the small sample size and the relatively stable condition of the participants.
Thank you for your valuable comments. As you rightly pointed out, acknowledging the study's limitations is essential. We have added a dedicated limitations section in the Discussion, where we address the sample size and the relatively stable condition of the participants. We appreciate your thoughtful suggestions and hope this revision enhances the clarity and balance of our discussion.
o The manuscript includes ethics and data availability statements are adequate.
Thank you for your kind comments. We appreciate your feedback and are pleased to hear that the ethics and data availability statements are adequate.
o Finally, I believe this is a great research and should be accepted for publication after minor revisions.
Thank you for your kind and encouraging comments. We truly appreciate your time and effort in reviewing our manuscript and providing valuable feedback. Your insights have been instrumental in refining our work, and we are grateful for your support.
Reviewer 2 Report
Comments and Suggestions for Authors
The submitted manuscript examines the use of the humanoid robot RoBoHoN in diabetes care consultations as a potential solution to address the global shortage of diabetes care professionals. The study describes the development of a robot-assisted consultation programme and analyses its feasibility through patient feedback. The findings suggest that robot-assisted consultations can elicit meaningful patient narratives and offer an alternative to human specialists.
The introduction highlights the global burden of diabetes but does not fully engage with recent studies on robot-assisted healthcare. Several recent publications from 2023–2024 discuss advancements in this field. Incorporating these would improve the contextual grounding and clarify the study’s contribution. The small sample size and reliance on a single operator limit generalisability, and these limitations require further discussion. While the methodology is clear, the analysis would benefit from additional statistical techniques or validation of the Likert-scale questionnaire to enhance reliability.
The results align with the study objectives, but some tables are overly detailed and could be summarised. The discussion interprets the findings effectively but overstates claims about scalability and utility in less experienced settings. These conclusions should be reframed to reflect the study’s preliminary nature. Streamlining the introduction and discussion would improve clarity and focus. Consistency in British English usage should also be ensured.
Comments on the Quality of English LanguageThe English in the manuscript is generally clear and well-structured, but it can be improved for conciseness and consistency. Sections such as the introduction and discussion are verbose and should be streamlined. British English should be used consistently throughout the text. Ensuring technical terminology is precise and consistent would further enhance readability.
Author Response
To Reviewer #2:
Comments and Suggestions for Authors
The submitted manuscript examines the use of the humanoid robot RoBoHoN in diabetes care consultations as a potential solution to address the global shortage of diabetes care professionals. The study describes the development of a robot-assisted consultation programme and analyses its feasibility through patient feedback. The findings suggest that robot-assisted consultations can elicit meaningful patient narratives and offer an alternative to human specialists.
Thank you very much for taking the time to review our manuscript and provide valuable feedback. We sincerely appreciate your thoughtful comments and insights.
The introduction highlights the global burden of diabetes but does not fully engage with recent studies on robot-assisted healthcare. Several recent publications from 2023–2024 discuss advancements in this field. Incorporating these would improve the contextual grounding and clarify the study's contribution.
Thank you for your thoughtful comments. In response to your suggestion, we have incorporated two recent studies from 2024 to provide a more comprehensive context for our research and clarify its contribution to the field. We appreciate your valuable feedback and hope these additions enhance our introduction's clarity and relevance. Please let us know if further revisions are needed.
The small sample size and reliance on a single operator limit generalisability, and these limitations require further discussion.
Thank you for your insightful comments. As you suggested, we have added details on sample size determination in the Methods section and clarified in the Discussion that, while the sample size of 32 is not large, it was set to ensure statistical validity while avoiding an excessively large sample that could yield statistically significant but clinically insignificant results. We appreciate your careful review and would appreciate your feedback on these revisions.
While the methodology is clear, the analysis would benefit from additional statistical techniques or validation of the Likert-scale questionnaire to enhance reliability.
Thank you for your thoughtful comments. As you pointed out, the statistical analysis section lacked sufficient detail. We have now expanded this section with additional information and included details on the validation of the Likert-scale questionnaire to enhance the reliability of our findings. We appreciate your careful review and would be grateful for your feedback on these revisions.
The results align with the study objectives, but some tables are overly detailed and could be summarised.
Thank you for your thoughtful comments. As you noted, some figures and tables contained excessive detail. To improve clarity and conciseness, we have revised and condensed them. The table displaying Japanese robot messages has been moved to the supplementary material. We appreciate your careful review and valuable feedback.
The discussion interprets the findings effectively but overstates claims about scalability and utility in less experienced settings. These conclusions should be reframed to reflect the study's preliminary nature.
Thank you for your insightful comments. We acknowledge that some statements may have overstated the implications of our findings. In response, we have revised the conclusions and corresponding abstract to more accurately reflect the preliminary nature of the study, emphasizing the potential for robot-assisted consultations to support, rather than replace, human experts. We appreciate your feedback and would appreciate your review of these modifications.
Streamlining the introduction and discussion would improve clarity and focus.
Thank you for your valuable comments. As you pointed out, the introduction and discussion sections were somewhat lengthy and could be more concise. We have revised them to enhance clarity and focus, making the text more straightforward and accessible. We appreciate your review of these modifications and look forward to your feedback.
Consistency in British English usage should also be ensured.
Thank you for your valuable comments. As you suggested, we have carefully reviewed and corrected any inconsistencies in British English usage. Additionally, if necessary, we will utilize MDPI's English editing service. We appreciate your careful review and constructive feedback.
Comments on the Quality of English Language
The English in the manuscript is generally clear and well-structured, but it can be improved for conciseness and consistency. Sections such as the introduction and discussion are verbose and should be streamlined. British English should be used consistently throughout the text. Ensuring technical terminology is precise and consistent would further enhance readability.
Thank you for your thoughtful comments. As you suggested, we have revised the introduction and discussion sections to enhance clarity and conciseness. We have also ensured consistent use of British English throughout the manuscript. Additionally, if needed, we will seek MDPI's professional English editing service. We sincerely appreciate your valuable feedback.
Thank you for taking the time to review our manuscript. With your thoughtful comments, our work has been refined.
Round 2
Reviewer 2 Report
Comments and Suggestions for Authors
The authors have addressed the key concerns from the initial review, notably by incorporating recent literature, refining the statistical analysis, and improving clarity in the discussion. The introduction provides relevant background, but the discussion would benefit from acknowledging financial barriers to robotic adoption as a potential limitation. This addition would strengthen the paper’s practical implications by recognising cost constraints as a factor in scalability. Additionally, while the study design is appropriate, further clarification on sample size determination and potential biases would improve methodological transparency.
Author Response
To Reviwer#2
The authors have addressed the key concerns from the initial review, notably by incorporating recent literature, refining the statistical analysis, and improving clarity in the discussion.
Thank you for your quick and thoughtful comments. We appreciate them
The introduction provides relevant background, but the discussion would benefit from acknowledging financial barriers to robotic adoption as a potential limitation. This addition would strengthen the paper's practical implications by recognising cost constraints as a factor in scalability.
Thank you for your valuable comments. To address your suggestion, we have discussed the financial feasibility of robotic implementation at the end of Section 4.2 and included two additional references. We would greatly appreciate your review of these revisions.
Additionally, while the study design is appropriate, further clarification on sample size determination and potential biases would improve methodological transparency.
Thank you for your thoughtful comments. We have added details on the estimation method used in applying Lehr's formula for sample size determination. Additionally, we have addressed potential biases in the study population by providing further clarification in the "Study Participants" section of the Methods. We would appreciate it if you could review these revisions.